# Bone Regeneration by Multichannel Cylindrical Granular Bone Substitute for Regeneration of Bone in Cases of Tumor, Fracture, and Arthroplasty

**DOI:** 10.3390/ijerph19148228

**Published:** 2022-07-06

**Authors:** Ki-Jin Jung, Swapan Kumar Sarkar, Woo-Jong Kim, Bo-Ram Kim, Jong-Seok Park, Byong-Taek Lee

**Affiliations:** 1Department of Orthopedic Surgery, Soonchunhyang University Cheonan Hospital, Cheonan 31151, Korea; c89546@schmc.ac.kr (K.-J.J.); 89489@schmc.ac.kr (W.-J.K.); jsparksch@schmc.ac.kr (J.-S.P.); 2Institute of Tissue Regeneration, College of Medicine, Soonchunhyang University, Cheonan 31538, Korea; swapan97@gmail.com (S.K.S.); sk7250@naver.com (B.-R.K.); 3Department of Regenerative Medicine, College of Medicine, Soonchunhyang University, Cheonan 31538, Korea

**Keywords:** granular bone substitutes, calcium phosphate, bone regeneration, bone defects

## Abstract

In orthopedics, a number of synthetic bone substitutes are being used for the repair and regeneration of damaged or diseased bone. The nature of the bone substitutes determines the clinical outcome and its application for a range of orthopedic clinical conditions. In this study, we aimed to demonstrate the possible applications of multichannel granular bone substitutes in different types of orthopedic clinical conditions, including bone tumor, fracture, and bone defect with arthroplasty. A clinical investigation on a single patient for every specific type of disease was performed, and patient outcome was evaluated by physical and radiographic observation. Brief physical characterization of the granular bone substitute and in vivo animal model investigation were presented for a comprehensive understanding of the physical characteristics of the granules and of the performance of the bone substitute in a physiological environment, respectively. In all cases, the bone substitute stabilized the bone defect without any complications, and the defect regenerated slowly during the postoperative period. Gradual filling of the defect with the newly regenerated bone was confirmed by radiographic findings, and no adverse effects, such as osteolysis, graft dispersion, and non-union, were observed. Homogeneous bone formation was observed throughout the defect area, showing a three-dimensional bone regeneration. High-strength multichannel granules could be employed as versatile bone substitutes for the treatment of a wide range of orthopedic conditions.

## 1. Introduction

A large proportion of the aged population has become increasingly vulnerable to musculoskeletal complications due to age-related health problems [1,2]. Various clinical conditions, which could be due to trauma, pathologic cause, and prior surgical intervention, may arise in this age group. Consequently, the demand for repair or regeneration of damaged or diseased skeletal tissue has also increased. To replace damaged hard tissue and to provide mechanical support with the augmentation of natural bone regeneration and healing process, bone graft substitutes are needed, which may be sourced from a biological origin (i.e., autotransplantation, allotransplantation, or xenotransplantation) or synthetically produced (i.e., made from bioceramics, bioglass, or biopolymers) [3,4,5,6,7]. Bone substitutes have emerged as an innovative medical solution for cavity filling in bone defects or for the repair of fractured bone. Although autotransplantation is regarded as the gold standard in orthopedic complications, the procedure is associated with limited availability and could result in donor site morbidity [8]. Moreover, both allotransplantation and xenotransplantation possess risk for disease infection and implant rejection [9,10]. Synthetic calcium-phosphate-based ceramic bone substitutes have been employed as bone graft substitutes for the past few decades [11,12,13]. Calcium phosphate is a mineral constituent of natural bone. In addition, hydroxyapatite (HAp), tricalcium phosphate (α/β-TCP), and biphasic calcium phosphate (BCP) have been used to create a porous scaffold that serves as an artificial bone-healing aid in orthopedics and dentistry, where early cellular regeneration of the defect site and mechanical stability of the implant are considered vital issues [14,15].

Some site-specific, application-specific, and performance-specific prerequisites for using a bone substitute material in hard tissue treatment exist [16,17,18]. The placement of a large bone graft in the defect site is also technically delicate because of restricted access or the need for limited invasion footprint. Readjustment or repositioning of the graft-loaded operated and treated bone with a single-size bone substitute may be a challenging task. Such considerations are addressed by granular bone substitutes, which could be easily applied to the defect site. Thus, granular bone substitutes may overcome the aforementioned complications. Various examples of the application of granular bone substitutes (processed bone or synthetically produced) in clinical practice have been reported [9,19,20,21,22]. However, fabricating porous bone graft with sufficient mechanical strength is challenging because the scaffold’s porosity compromises strength. To overcome this problem, multi-pass extrusion fabrication has been used to create calcium phosphate-based multichannel porous granules (MCG), which allows sufficient control of pore size, shape and mechanical properties. This technology also minimizes the inverse relationship between compressive strength and porosity, which is a limitation with many of the existing synthetic bone graft materials. A granular bone substitute could be used as an efficient filler, with ease of applicability and adequate conformity to the defect volume. However, optimum control of the interconnected porosity while maintaining strength is a persistent challenge for bone substitutes, especially for granular bone substitutes [23]. A cylindrical granule with multiple through channels could be a good choice, as the regular shape demonstrates strength and the through channels may facilitate easy cellular proliferation. Furthermore, inter-granular space provides additional porosity, which is thoroughly interconnected. We have previously reported the development of a cylindrical granular bone substitute, which has improved mechanical properties and holds promising potential for bone regeneration [24]. In this study, we aim to clinically investigate in detail the usability of multichannel granular bone substitutes for the treatment of a range of clinical conditions, including bone tumor, fracture, and bone defect with arthroplasty.

## 2. Materials and Methods

This retrospective study was approved by the institutional review board at Soonchunhyang University Hospital (SCHCA 2016-07-010). Informed consent was waived because of the retrospective nature of the study; nevertheless, all patients provided written informed consent. In addition, the authors confirm that all ongoing and related trials for this intervention are registered.

### 2.1. Granule Fabrication

BCP nanopowders (60 Hap + 40 β-TCP), with an average particle diameter of 70–100 nm, were used to fabricate the porous granules by the multi-pass extrusion process, as described previously [24]. Briefly, BCP/polymer shell was casted using a warm press, and a carbon/polymer core was extruded to be employed as fugitive pore former. The casted shell and extruded core were joined and extruded to produce a filament of approximately 3.5 mm in diameter. This first pass filament was subsequently used under different extrusion settings to create granules with different sizes and pore arrangements. The first pass filaments were cut into a fixed length and assembled in a configuration of six filaments encircling a single central filament. All filaments were enclosed by another shell and subsequently extruded together to make another filament of approximately 1 mm in diameter (the second pass filament). These second pass filaments were cut (1 mm length) to obtain the green stage granule that is ready for polymer removal and sintering. After heat treatment, the granules were obtained in their final form. For 2 mm and 3 mm granules, different numbers of first pass filaments were assembled and extruded with second pass filaments measuring 2 mm and 3 mm in diameter, respectively. Thereafter, the green granules were subjected to controlled burning-out and sintering process as usual. To remove the polymer, the granules were subjected to slow heating in a nitrogen atmosphere for 5 days, with temperature reaching 700 °C. After polymer removal, a second burning-out process was performed at 1000 °C for 2 h in an air atmosphere to burn out the pore-forming carbon and any other residual carbon obtained from polymer pyrolysis. Finally, a pressure-less sintering process at 1350 °C for 2 h in an air atmosphere was performed to obtain the BCP granules with a porous structure. The granules in our study were provided by Innobone (Asan City, Korea); the company has received the Korea Food and Drug Administration approval for the clinical application of the multichannel granular bone substitutes with the brand name “Frabone”.

### 2.2. Morphology, Porosity, and Mechanical Strength

Optical images of the multichannel BCP granules were obtained for comparative morphological analysis. Granular bone substitutes had large macro pores and a well-sintered microstructure without any visible pores within the dense matrix. Due to the well densification and defined pore structure of the BCP granules, porosity was determined by the weight dimension method. Moreover, pore volume was calculated based on intrusion of water under a high ‘g’ load by centrifugation. An aggregate of around 1 cm^3^ of granules was employed (which is relevant to a small defect size) for percent porosity calculation using the intruded water volume. The porosity value obtained was the total porosity of the granule aggregates (combining both inter-granular and intra-granular porosity). Compressive strengths of the individual granules were determined using a universal testing machine (Unitech, R&B, Daejeon, Korea). The granules were tested with a 100 kg load cell at a crosshead speed of 1 mm min^−1^. To determine the failure load of bulk granule, a stainless-steel mold measuring 7.5 mm in diameter was made and compressive load was applied to a bulk aggregate of granules within the mold. The loose and unbound granules could only take load determined by the individual granule’s compressive strength. For the evaluation of all the properties, measurements were performed at least five times, and the average value was considered

### 2.3. In Vivo Characterization

To ascertain the bone regeneration ability of granular bone substitutes in an animal model, in vivo investigation was conducted by implanting 1 mm granular bone substitutes in a critical size defect of a rabbit femur for 4 and 8 weeks. The study design was submitted to and approved by the ethical committee of Soonchunhyang University for animal use. For each set of samples, *n* = 3 were used. Prior to the operation, the surgical area was shaved and the skin was washed with iodine solution. General anesthesia was induced using a combination of 1.3 mL ketamine (100 mg mL^−1^; Ketara, Yuhan, Seoul, Korea) and 0.2 mL xylazine (7 mg kg^−1^ body weight; Rompun, Bayer Korea, Seoul, Korea), which was administered intramuscularly. After exposure of the femoral head, a defect measuring 5 mm in diameter and 5 mm in depth was made using a trephine drill, which was subsequently filled with the specific type of granules (experimental group) or was left without any filling (negative control). After the surgery, the surgical sites were closed by suturing. Antibiotics (Baytril, Bayer Korea) and analgesics (Nobin, Bayer Korea) were administered intramuscularly for 3 days to prevent postsurgical infection and to control pain. The animals were sacrificed by intravenous administration of air under general anesthesia after the specified time span and the implanted femur head was extracted. Tissue sections from formalin-fixed paraffin-embedded tissue were extracted by ultramicrotome. The sections were histologically stained using hematoxylin and eosin (H&E) and were observed under a light microscope.

### 2.4. Clinical Surgical Intervention

In our investigation, three different types of clinical pathophysiological conditions of bone graft requiring surgical intervention for functional restoration of bone were investigated. Fracture, bone tumor, and bone defect with arthroplasty were treated with granular bone substitute. Patients who needed a bone graft procedure between January 2015 and February 2016 were enrolled in this study. While the follow-up period differed among the patients, it was at least >12 months. Patients between ages 15 and 80 years were included in this study. The exclusion criteria were as follows: patients with autoimmune disease, malignancy, or infectious disease and those who did not sign the informed consent form. Clinical and radiographic data were obtained in accordance with the regulations of the institutional review board at Soonchunhyang University Hospital.

Radiological data and patient condition were evaluated to determine the outcome of the application of granular bone substitute. The study progressed with the tracking of bone regeneration extent through radiological findings, and the clinical outcome, including general health status, pain status, patient satisfaction, and patient global perceived effect, were also assessed. The clinical outcomes were analyzed to relate with the specific features of the granular bone substitutes.

#### 2.4.1. Granular Bone Substitute for Bone Tumor

##### Fibrous Dysplasia

A 34-year-old female visited our outpatient department with local tenderness on the anterior aspect of the hip joint area. Radiographs and computed tomography (CT) scan were obtained and the results showed an oval-shaped cystic lesion in the femoral intertrochanteric area. Radiological assessment confirmed the diagnosis of fibrous dysplasia that requires surgical intervention. At 3 months after the occurrence of the initial symptom (i.e., pain), the operation was performed. A surgical window was made around the anterolateral aspect of the femur area, and the mass was found to contain fibrotic tissue and serous fluid. The mass was removed by curettage and was sent to a pathologist for evaluation. The bone void created after the lesion removal was filled with a 1 mm granular bone substitute to stabilize the defect area, and no additional fixation device was used for mechanical support. One week after surgery, the patient was discharged, and the pathological finding was fibrous dysplasia with cystic change. Follow-up radiographic examinations were performed to evaluate the progression of bone development. At 2 weeks, stitch out was performed, with good healing of the hip wound. After a month’s time, the patient recovered well with no complaints and returned to work.

At 1 year’s postoperative follow-up, the patient did not complain of any specific discomfort, and radiological findings showed no specific abnormalities in the surgically performed tumor site.

##### Simple Bone Cyst

A 51-year-old female visited our outpatient department with pain in the left hip for past 6 months. Complete physical examination at the outpatient department revealed local tenderness over the anterior aspect of the hip joint. Radiographs and CT showed a round-shaped cystic lesion measuring 4.0 × 3.0 × 3.0 cm in the femoral intertrochanteric area. Radiologist inferred that the lesion was a simple, hemorrhagic bone cyst, and a decision was taken for surgical intervention. The planned surgery involved curettage of the cyst with subsequent bone grafting. At 6 months after the onset of the initial pain, operation was performed. A surgical window was made around the anterolateral aspect of the femur area, and the mass was found to contain fibrotic tissue and serous fluid. The mass was removed by curettage and was sent to a pathologist for evaluation. After removal of the mass, granular bone substitute was applied to the defect site, and decision was taken for additional fixation for stabilization. Dynamic hip screw (Synthes, Oberdorf, Switzerland) was selected for the fixation.

A week after the surgery, the patient was discharged without any complications, and the pathologist defined the mass (4.0 × 3.0 × 3.0 cm) as a simple bone cyst with a hemorrhagic change, which was similar to the initial findings of the preoperative radiologic evaluation. A follow-up radiograph was obtained in the outpatient department at 2 weeks after the surgery. Moreover, at 2 weeks, stitch out was performed, with good healing of the hip wound. At 1 month postoperatively, the patient recovered well with no complaints and returned to work.

The radiograph showed a bony defect filled with granular bone substitutes with good maintenance. Radiological and clinical findings revealed that the granular bone substitutes promoted bone regeneration without any complications. At 3 months postoperatively, the patient did not complain of pain during examination in the outpatient department. At 2 years postoperative follow-up, the patient did not complain of any specific discomfort, and radiological findings showed no specific abnormalities in the surgically performed tumor site.

#### 2.4.2. Granular Bone Substitute for Bone Fracture

##### Fracture from Cystic Bone

A 15-year-old female slipped down and required emergency medical attention. She complained of severe pain and swelling on the left hip joint area was reported, and radiograph and CT scan found a fracture in the intertrochanteric area of the femur. The fracture was not simple as a large bone cyst was noted; thus, it was a pathologic fracture. Skeletal traction was applied for fracture stabilization, and operation was scheduled. After 2 days from the initial trauma, the operation was performed. The fracture was fixed with a long proximal femoral nail antirotation (PFNA) system (Synthes, Oberdorf, Switzerland). The bone defect was extremely large. We assumed that fracture healing is difficult and treatment requires not only fixation but also massive bone grafting. Granular bone substitute was applied in the fracture region for stabilization.

At 10 days after the surgery, the patient could walk using a crutch and returned home without any complications. At 3 months postoperatively, the patient visited our outpatient department and a follow-up radiograph was obtained. The radiograph showed that the initial fracture was stabilized by the internal fixation device and the bony defect was filled with the granular bone substitutes well, and revealed that the granular bone substitutes promoted bone regeneration without complications. At 3 years postoperative follow-up, the patient did not complain of any specific discomfort, and radiological findings showed union at the fracture site.

#### 2.4.3. Granular Bone Substitutes for Bone Defect with Arthroplasty

##### Primary Arthroplasty

A 66-year-old female visited our outpatient department and complained of bilateral hip pain. The patient had been suffering from bilateral hip pain for 2 years. Physical examination revealed local tenderness over the anterior aspect of the bilateral hip joint, and bilateral Patrick’s test was positive. In addition, radiographs, CT scan, and magnetic resonance image (MRI) were obtained, which showed avascular necrosis on the bilateral femoral head and a 4 × 3 cm bony defect at the right anterosuperior aspect of the acetabulum. The findings revealed necrosis and consequent defect of the femoral head, and total hip replacement surgery was planned. 

Operation was performed by approaching the posterior aspect of the hip joint using the modified Gibson approach. We found that the femoral head and acetabular area were damaged and we replaced them with an artificial joint. After device insertion, a granular bone substitute was applied to the bony defect on the acetabulum for stabilization.

One week after surgery, the patient could walk using crutches and returned home without any complications, and at 2 weeks, stitch out was performed, with good healing of the hip wound. At 3 months postoperatively, the patient visited our outpatient department and a follow-up radiograph was obtained. At 6 years postoperative follow-up, the patient complained mild hip pain, but it was not severe. Additionally, there were no specific abnormalities such as osteolysis in the defect site around arthroplasty.

## 3. Results

### 3.1. Physical Characterization of the Multichannel Granular Bone Substitutes

Figure 1 shows the cylindrical-shaped granular multichannel bone substitutes. Three types of granules with increasing size were employed, with a diameter of 1 mm, 2 mm, and 3 mm; the height of the granules was 1 mm, 1.2 mm, and 1.4 mm, respectively. The granules differed with respect to the number of pores and size of pores and were designed to have 7, 14, and 16 channels along the axial direction of the cylindrical granule [25]. Each granule possessed seven channels extending from one end to another. The channel openings were extremely rough and irregular. No severe processing defects, such as cracks, crevices, or lack of sintering, were observed in the granules. The inner surface of the channels was extremely rough with unidirectional ridges that ran parallel to the channel length (Figure 1a,c).

The porosity and compressive strength of the individual granules are presented in Figure 2. The pore size of the 1 mm, 2 mm, and 3 mm granules was 150 ± 15, 190 ± 22, and 320 ± 31 μm, respectively (Figure 2a), and details can be found at our previous publication [25]. Variable pore size and frame of the granules had a direct influence on their mechanical properties. The compressive strength of the granules is higher with a higher diameter (Figure 2b). Failure load of granule aggregates was measured separately to understand the actual load-bearing capability of the applied granules before they get partially crushed [25]. As the granules are unbound with each other, applied load is directly experienced by each granule rather than by the bulk aggregate as a single body. This prohibits the determination of the true compressive strength of the bulk, and thus, instead, we measured the failure load directly with a granule aggregate in a mold (height, 7.5 mm; diameter, 7.5 mm).

### 3.2. In Vivo Study to Evaluate Early Bone Regeneration Performance

To assess the overall filling and healing of the defect site and to evaluate early bone regeneration performance with the granular bone substitute implanted in a large-size bone defect, a rabbit model was employed in this study. Defect filling was found to be significantly higher in the animals with granular bone substitutes than in the control as shown in the descriptive histological analysis (Figure 3) of tissue sections at the implant sites. Quantitative percentage of bone volume fraction (BV/TV%) results of cylindrical TCP and BCP calculated from micro-CT data after 1 week,1 month, and 6 months of implantation are reported in our previous publication [25]. At 6 months after implantation, bone volume fraction for BCP scaffolds was 30 ± 1.1%. Formation of both soft and hard calluses was observed within 8 weeks based on the H&E staining of tissue slices. The result demonstrates a distinct superior performance of the granular bone substitutes in general. Moreover, a gradual increase in bone formation and development of new bone tissue towards the succeeding sampling dates was evident in accordance with previous published work [25]. At subsequent stages, the defect was observed to be filled out volumetrically, exhibiting superior neo-bone formation. Within 4 weeks, the entire defect was filled with new bone with BCP. Bone formation was observed both inside the channel and in the inter-granular space. Thickening of the interface region and fusion of the new bone with the original bone was ensured. Proper filling and enhanced stability of the defect, and these results are reproducible in accordance with our preliminary work [25].

Micro-blood vessel formation was observed at the center of the channels. This blood vessel formation is very desirable for the overall healing mechanism, both initially during the new bone formation and subsequently during the remodeling phase.

### 3.3. Clinical Results

In all the patients, the multichannel granule ensured good defect filling and stabilization with a complication-free outcome. All the patients were able to return to regular movements within 2 months postoperatively. The case histories of all the patients examined with different clinical disorders are summarized in Table 1.

#### 3.3.1. Bone Tumor

##### Fibrous Dysplasia

Figure 4a,b show homogeneous loss of the normal trabecular pattern based on the radiographs in the metaphysical region of the femur, with a clearly defined darker contrast appearance due to fibrous dysplasia. The medullar bone was largely replaced with fibrous tissue. Figure 4a,b,i confirm the existence of a fibrous tissue around the lesser trochanter region. Postoperative radiographs (Figure 4c,d) showed complete removal of the tissue lesion and refilling with bone substitutes. Radiographs obtained after 1 (Figure 4e,f) and 12 (Figure 4g,h) months revealed gradual recovery of the defect with bone regeneration. A homogeneous bone formation was observed throughout the whole area with dysplasia. The CT image (Figure 4j) obtained at 12 months after surgery showed an extensive bone formation and regeneration in the area that previously had fibrous dysplasia. The compact bone lining between the host bone and implant, which was visible in the radiograph, progressively faded away, thereby confirming the fusion of the original bone and the newly regenerated bone. The patient did not complain of any pain or discomfort after 1 year of treatment.

##### Simple Bone Cyst

Bone cyst removal and the subsequent granular bone substitute application were performed in a similar fashion. The cyst at the neck of the femur (Figure 5a) was diagnosed by preoperative radiographic observation. MRI showed a dark circle in the femur neck (indicating cyst formation) (Figure 5b). After curettage, a dynamic hip screw system was used to stabilize and reinforce the femur neck, which had become vulnerable to fracture due to loading. Granular bone substitutes were applied to fill the cavity formed after the cyst removal. Figure 5c,d show the postoperative radiographs demonstrating complete filling of the void with bone substitutes. The granules could be seen in the defect area as inhomogeneous contrasts. After 2 years, enhanced homogeneity was observed, confirming full regeneration of the defect site with new bone (Figure 5g,h). A distinct cortical line appeared after 2 years postoperatively (Figure 5g,h); the line ascertained the existence of bone-remodeling process.

#### 3.3.2. Fracture

##### Bone Fracture with Cyst Formation

Figure 6a,b show the MRI of the fractured bone in both the coronal and axial planes. A compound fracture is evident in the MRI. The cyst formation complicated the resulting fracture (Figure 7a,b). Postoperative radiographs after stabilizing the femur by PFNA system and filling the bone defect with granular bone substitutes are shown in Figure 7c,d. At 5 months postoperatively, a significant progression in bone regeneration was observed along with a homogeneous and continuous bony structure. A substantial change in bone shape in association with additional growth down the fracture region, with stabilization of the contour fattening around the fracture zone, was observed (Figure 7g,h).

#### 3.3.3. Bone Defect with Arthroplasty

##### Primary Arthroplasty

The radiographs in Figure 8a,b show avascular necrosis of the bilateral femoral head and 4 × 3 cm bony defect at the right anterosuperior aspect of the acetabulum. The necrosis extended and the femur head was damaged. Total hip replacement surgery was performed with an artificial hip joint, and the defect area with bone necrosis was filled with a granular bone substitute (Figure 8c,d). Bone regeneration was evident at 1 month postoperatively (Figure 8e,f). After 6 years postoperatively, bone formation significantly increased as indicated by the homogeneous contrast in Figure 8g,h. Host bone and newly regenerated bone integration was well maintained, except in some parts of the posterior acetabular region (shown as a shadowy line). Moreover, the radiographs show a well-maintained prosthetic device and complete filling of bony defect with granular bone substitutes. Promotion of bone regeneration by the granular bone substitutes without any complications was also observed.

## 4. Discussion

Bone regeneration is a complex process, which involves the coordinated function of bone-forming osteoblasts and bone-resorbing osteoclasts. The actions of osteoblasts and osteoclasts are mediated and regulated by a complex environment of biochemical agents present at the remodeling site. In the primary stage, the cells perform their task by being active on the surface. Bone regeneration progresses with layer formation through the extracellular matrix deposition and subsequent crystallization of hydroxyapatite nanoparticles. This could explain why a porous microstructure is an essential requisite feature for all high-performance implant materials; the porous space provides the necessary surface for the osteogenic cells to mediate their activity. Moreover, interconnected porosity of an implant enables volumetric occupation of osteogenic cells from all sides of the bone defect, thereby ensuring overall ossification of the implant material in a much-reduced time. Such a feature is also a key requirement for the onset of blood vessel and nerve system distribution, which is integral for the regenerated and more mature and structurally configured remodeled bone. In addition, the primary architecture of cortical bone is composed of a cluster of unidirectional concentric lamellar unit with a central channel called osteon. The porous structure required to induce new bone formation is preferentially a unidirectional channel that could mimic the original bone micro-architecture. This feature is an inherent structural aspect of the granular bone substitutes used in this study, which could mediate guided bone regeneration in a preferential orientation. The unidirectional channel in a regular geometric structure could also be translated into higher mechanical properties of the granules compared to random porous bone substitute scaffolds. This implies that a micro-channeled porous architecture could be a better option for the bone substitute microstructure. It is hypothesized that, compared to conventional three-dimensional porous bone substitutes, the channeled and granular bone substitutes have better possibilities for bone remodeling. A cylindrical granule with multiple channels may be a good choice. The regular cylindrical shape may indicate geometric stability that renders strength to the granules, and multiple channels may facilitate easy proliferation of cells.

In our study, the granular bone substitutes were successfully applied in three different bone defects, and in each case, the defect site was fully stabilized after the application of the bone substitutes. The granules were confined within the defect site, and no dispersion of the granules to the surrounding tissue area was observed. This was true in both the confined defect (not exposed), as in the case of fibrous dysplasia, and in the open defect (exposed), such as bone fracture. Granular confinement is important not only because it is relevant for quick bone regeneration but also because it could provide an early sign or indication of any refracture in the newly regenerated bone. In all the cases presented in this study, the radiographs did not show any anomalous contrast of the granular presence around the defect site. However, it is not possible to identify any broken particulates from the granules in the radiographs. Wear debris from heterogeneous implant materials is capable of inducing osteolysis, leading to secondary bone trauma. From our observation, such wear debris did not occur in the cases with multichannel granular bone substitutes.

Graft union with the host bone is important for osteointegration and the determination of the final fate of the bone with defect. In our study, in confined graft application, such as that in dysplasia (Figure 4) and bone cyst (Figure 5), gradual union between the graft and host bone interface was observed in the postoperative radiographs. Densification increased as time passed by. In unconfined defects (such as fracture of the femoral head; Figure 7), the osteointegration and restoration of the functional bone with structural contour was driven by local physiological influence and the bone graft was vital for the regeneration process. As can be seen in Figure 7, structural configurations of the formed bone were slightly altered as compared to those of a standard femur bone, and bony growth was associated with the spatial distribution of the applied granules. However, no disadvantageous outcome in the local anatomy and the patient outcome were observed, and the results were satisfactory.

Bone complications associated with implant application needs a rigorous assessment. Especially, arthroplasty requires a safe and clean scaffold without the risk of any premature fracture debris from the implant. The implant must not induce any infection, irritation, or loosening. Nonetheless, given the dynamic nature of the hip joint and the articulation of the artificial hip joint in the femur head and acetabular cup, a prevalent risk of fine debris formation that could trigger debilitating consequences, such as loosening and osteolysis, already exists. Thus, the multichannel granular bone substitutes may come especially useful because of its high strength structure and defect-free surface finish, which in turn could greatly minimize the risk of debris formation during sudden or asymmetric loading conditions. In case of bone grafting, where direct load bearing at some point is anticipated prior to full regeneration of bone, it is especially important as it could be a bone defect with arthroplasty. Even the settlement of granules could lead to additional trauma.

Radiography alone could not provide the whole picture of osteointegration. Bone in-growth could not be directly confirmed solely by radiological data. Neovascularization and extension of the vascular network are vital for bone regeneration and could determine the extent of bone regeneration. They can be ascertained by bone biopsy and histological analysis; however, they are highly restricted because of ethical reasons and thus were not performed in our study. Vascularization affects the pace of regeneration within the region of defect in a three-dimensional manner. Surface attachment of osteogenic cells, cell proliferation with layer formation, and stratification of the layers occur concurrently with blood vessel formation. Thus, bone regeneration must be accompanied with vascularization. All these aspects signify the importance of porosity, especially interconnected porosity, in bone substitutes. The multichannel granular bone substitutes are thus an ideal choice for the advancement of regeneration from the graft–host bone interface to the interior of the bone graft material. The intra-granular channels could ensure the passage through the granules, while the inter-granular tortuous porosity could ensure unobstructed passage of blood vessel from one end of the host bone to the other end through the applied bone substitute mass.

Conventional block-type single-piece ceramic bone substitutes are generally lacking high mechanical strength and high structural integrity because of the defects inherent to ceramic processing. A precise control of pore size and frame thickness is also not guaranteed in most of the bone substitutes. However, these two aspects are vital for a predictable performance, especially for the biodegradation of the scaffold. Biodegradation, which progresses with both physical and cellular activities, relies on the end-to-end resorption of the scaffold frame, and a regular and homogeneous distribution of the scaffold frame could translate to a more predictable outcome. All the employed granular bone substitutes in our study have a defined geometry in terms of granule size, length, pore diameter, and frame thickness. Thus, the degradation behavior would be much more predictable, despite the presence of other external factors, such as patient’s condition, age, and different individuals that strongly influence degradation. 

The size variation of the granules enables the surgeons to select the appropriate granules according to the size, geometry, and location of the bone defect. In addition, the highly sintered bone substitute is highly radiopaque; thus, it is easy to identify the postoperative condition of the defect zone with X-ray along with the extent of bone regeneration and healing. However, as a non-gold-standard option, the granular bone substitutes investigated in this study showed some drawbacks, too. Due to the small size of the granules, special care and attention are needed during the placement of the granule within the defect. Unlike the block-type bone substitutes, the granules could not be placed using forceps in a single step; a spoon is used instead, which requires additional care. Splashing may occur, which may require repositioning of the granules using forceps. A better way is to use a canola, which enables placement of the granules in a less invasive manner without splashing. However, jamming of the canola conduit with mechanical interlocking of the granules may sometimes occur. The use of a secondary graft material, such as demineralized bone matrix, could greatly enhance the operation procedure and thus could improve bone regeneration performance. Splashing is more evident in bone defects with a shallow depth and is largely reduced in defects with appreciable depth. Hence, special care is needed for the application of granular bone substitutes in bone defects with a shallow depth.

The overall efficacy of multichannel granular bone substitutes was excellent, and the intra-granular and inter-granular porosity ensured good bone healing. The application of the granular bone substitutes was easy and less invasive because of its granularity and small size. The high strength of the granules translated into stable defect filling without any secondary trauma, thereby leading to early recovery from surgery. Hence, the multichannel granular bone substitutes are considered a promising choice for the treatment and healing of various orthopedic clinical conditions.

## 5. Conclusions

Multichannel granular bone substitutes are a versatile choice for the treatment of various bone defects. Different kinds of bone trauma associated with the femur and hip, which were either pathological or physical, were successfully healed with the multichannel granular bone substitutes. In cases of bone tumor, fracture, and bone defect with arthroplasty, the granular bone substitutes resulted in stabilization of the defect, and all the patients exhibited recovery within a short time. Successful bone regeneration was observed in all the patients without any complications. Based on the radiological findings and patient’s quality of life, the patients with different orthopedic conditions have recovered.

## Figures and Tables

**Figure 1 ijerph-19-08228-f001:**
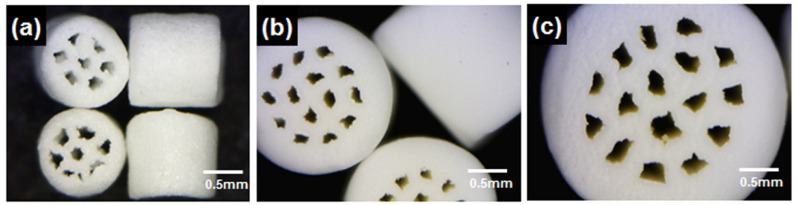
Optical micrographs of different size multichannel granular bone substitutes. (**a**) 1 mm diameter, (**b**) 2 mm diameter and (**c**) 3 mm diameter.

**Figure 2 ijerph-19-08228-f002:**
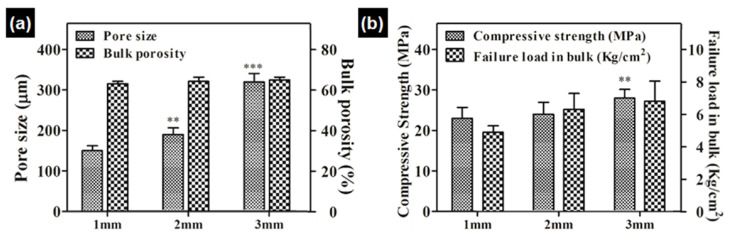
Physical and mechanical characterization of the granular bone substitute (**a**) pore size, (**b**) compressive strength. ** ≤0.01, *** ≤0.001.

**Figure 3 ijerph-19-08228-f003:**
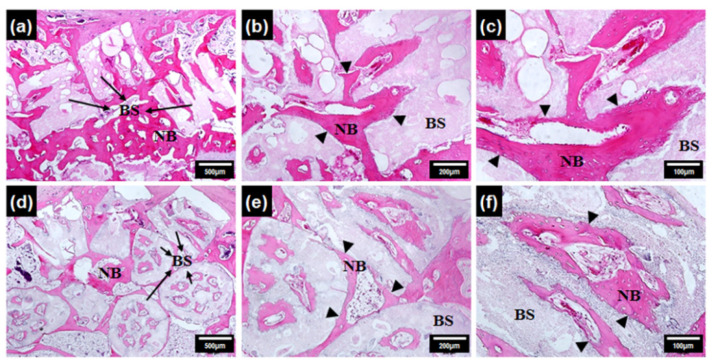
H&E-stained tissue section of granular bone substitute implanted bone areas. (**a**–**c**) Bone formation after 1 month of implantation with 1 mm granule. (**d**–**f**) bone formation after 2 months of implantation. Legend: BS (Bone substitute), NB (New Bone).

**Figure 4 ijerph-19-08228-f004:**
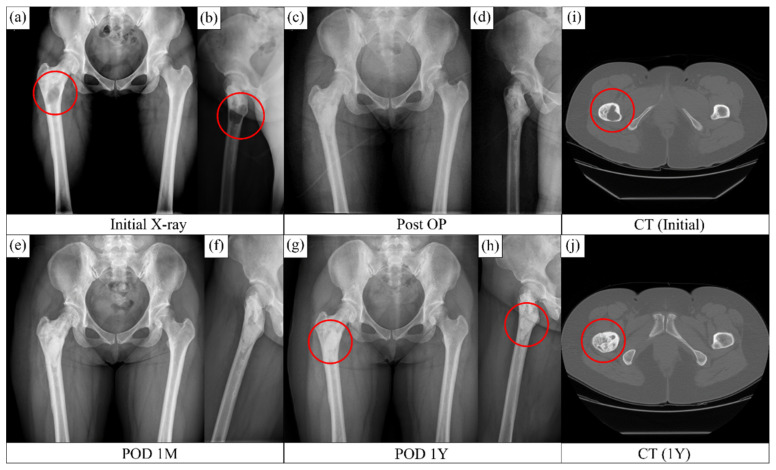
Bone regeneration in the bone void due to the osteofibrous dysplasia. (**a**,**b**) Preop condition, (**c**,**d**) postop condition, (**e**,**f**) after one month of operation and (**g**,**h**) after 12 months of operation. CT images of the (**i**) tissue lesion of the dysplasia before operation and (**j**) regenerated bone after 12 months postop. The red circles indicate implanted defect zone.

**Figure 5 ijerph-19-08228-f005:**
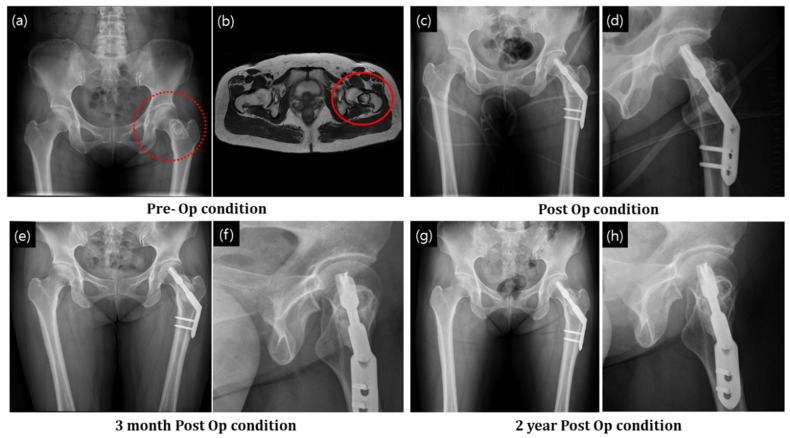
Simple bone cyst in the intertrochanter region of the femur was removed and granular bone substitute used for bone regeneration. (**a**,**b**) X-ray and MRI diagnosis of the bone cyst, (**c**,**d**) postop condition with application of granular bone substitute and support implant, (**e**,**f**) after 3 months of operation and (**g**,**h**) after 3 years of operation. The red circles indicate implanted defect zone.

**Figure 6 ijerph-19-08228-f006:**
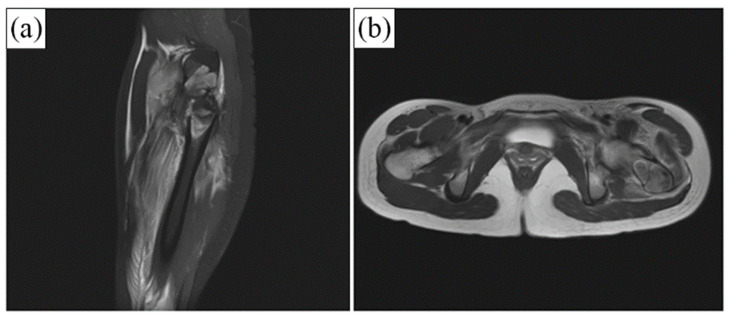
MRI of simple bone cyst formation in the intertrochanter region of the femur causing a compound fracture. (**a**) MRI of the fractured bone in the coronal plane. (**b**) MRI of the fractured bone in the axial plane.

**Figure 7 ijerph-19-08228-f007:**
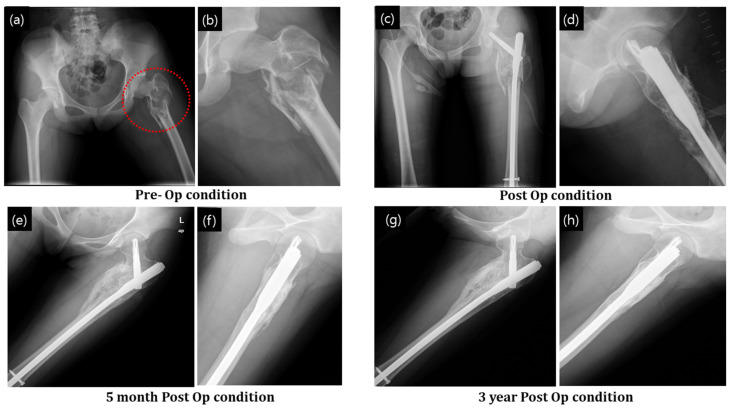
Fracture of intertrochanter region of the femur after a cyst formation was treated with granular bone substitute. (**a**,**b**) X-ray of the bone fracture, (**c**,**d**) postop condition with application of granular bone substitute and support implant, (**e**,**f**) after 5 months of operation and (**g**,**h**) after 3 years of operation. Red circle indicate defect zone with fracture.

**Figure 8 ijerph-19-08228-f008:**
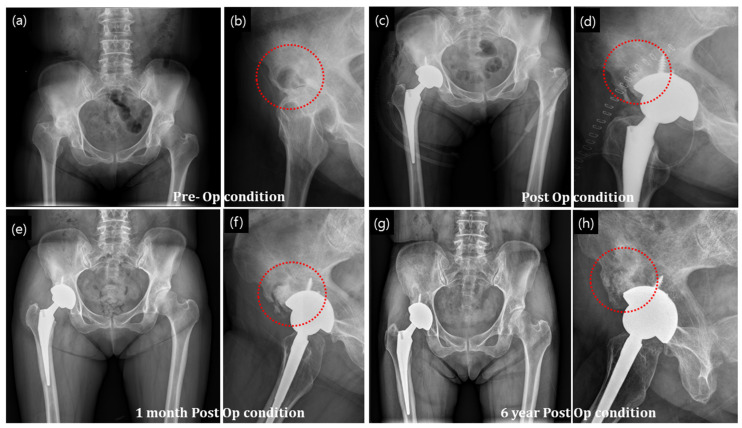
Primary arthroplasty of the hip joint treated with granular bone substitute for restoration of new bone in the necrossed bone (**a**,**b**) X-ray of the bone with necrosis, (**c**,**d**) postop condition with application of granular bone substitute around the hip joint implant, (**e**,**f**) new bone formation at the granular bone graft site after one months of operation and (**g**,**h**) after 6 years of operation condition. The red circles indicate implanted defect zone.

**Table 1 ijerph-19-08228-t001:** Patients case histories.

Case	Age	Follow Up Period (m)	Union	Harris Hip Score (Improvement)
Fibrous dysplasia	34	18	Complete	Excellent
Bone cyst	51	24	Complete	Excellent
Pathologic fracture	15	36	Complete	Excellent
Primary athroplasty	66	72	Complete	Excellent

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
