# Peer review of "Bone Regeneration by Multichannel Cylindrical Granular Bone Substitute for Regeneration of Bone in Cases of Tumor, Fracture, and Arthroplasty"

_ijerph, 2022, doi:10.3390/ijerph19148228_

Round 1
Reviewer 1 Report
Thank authors for their answers. But unfortunately, in answer to my 2nd question I did not recieve satisfying answer. Histomorphometry and microCT analysis are two completely different types of quantification analysis, one is 2D and second one is 3D. I am aware that you have quantified it by microCT in your previous work, but in this article you are stating that you have used histomorphometry analysis but there is no quantification, just histological slides. Histomorphometry is refering to the quantification of distinctive part of histological slide (e.g. bone) by quantifiyng specific color (pixels). If you are not able to do histomorphometry, then rather write that you used descriptive histological analysis (section 3.2., 4th row).
Reviewer 2 Report
The work has been revised in accordance with the reviewer suggestion, although in a not so deep manner. The introduction could be improved by including a more deep focus on the novelty introduced by the authors. Regarding morphological and microstructural analysis, the authors could refer in the text to the published manuscript, thus making complete the work including missing data. The same comment could be extended to the in vivo analysis.
Author Response
Please see attachment

This manuscript is a resubmission of an earlier submission. The following is a list of the peer review reports and author responses from that submission.
Round 1
Reviewer 1 Report
The manuscript submitted to International Journal of Enviromental Research and Public Health entitled “Clinical applications of Multichannel Cylindrical Granular Bone Substitutes for Bone Regeneration in Tumor, Fracture, and Arthroplasty” is an article in which authors are presenting clinical cases of fibrous dysplasia, bone cyst, fracture and arthroplasty treated with biocompatible multichannel cylindrical granules based on calcium phosphate as a substitution for golden standard for treating aforementioned indications, autograft. As a reviewer, I find this article very interesting, well structured and written, but some changes and revisions are needed.
First of all, in Materials & Methods section, authors are stating that granules used are made from biphasic calcium phosphate which consists of HAp and TCP, but there is no information about the final ratio of HAp and TCP? This is important information, because it is known that HAp and TCP have different resorption dynamics – HAp is more resilient to resorption in comparison to TCP. Knowing that, it is important to discuss for which indication different composition is preferable. For example, for nonunion fractures, TCP or BCP with higher percentage of TCP would preferable in order to achieve full rebridgment and reconstitution of the segmental bone defect. I would suggest to cite article by Pecin et al. „A novel autologous bone graft substitute containing rhBMP6 in autologous blood coagulum with synthetic ceramics for reconstruction of a large humerus segmental gunshot defect in a dog: The first veterinary patient to receive a novel osteoinductive therapy“ (doi: 10.1016/j.bonr.2021.100759).
As a second major revision is directed to the animal experiments. First of all, sample number (number per experimental group) used was not mentioned in M&M section. Authors are stating that „defect filling was significantly higher in the animals with granular bone substitutes than in the control as shown in histomorphometry analysis“, but it is not elaborated in M&M how histomorphometry was performed and how was newly formed bone quantified? Also, microCT analyis would be an great option for a 3D quantification of newly formed bone. If there is no option for microCT analysis, I would suggest 2D histomorphometry analysis for newly formed bone described in Stokovic et al. „Autologous bone graft substitute containing rhBMP6 within autologous blood coagulum and synthetic ceramics of different particle size determines the quantity and structural pattern of bone formed in a rat subcutaneous assay“ (doi: 10.1016/j.bone.2020.115654).
Author Response
Query1: First of all, in Materials & Methods section, authors are stating that granules used are made from biphasic calcium phosphate which consists of HAp and TCP, but there is no information about the final ratio of HAp and TCP? This is important information, because it is known that HAp and TCP have different resorption dynamics – HAp is more resilient to resorption in comparison to TCP. Knowing that, it is important to discuss for which indication different composition is preferable. For example, for nonunion fractures, TCP or BCP with higher percentage of TCP would preferable in order to achieve full rebridgment and reconstitution of the segmental bone defect. I would suggest to cite article by Pecin et al. „A novel autologous bone graft substitute containing rhBMP6 in autologous blood coagulum with synthetic ceramics for reconstruction of a large humerus segmental gunshot defect in a dog: The first veterinary patient to receive a novel osteoinductive therapy“ (doi: 10.1016/j.bonr.2021.100759).
Answer: Authors are thankful to the reviewers for their critical observation. We add the information in the M&M part. BCP consisted of 60 HAp and 40TCP samples were used for this investigation. (2nd paragraph of Material and Methods) We appreciate your comments about article about BMP. However we are failed to find any relevance with the suggested research article, therefore we are not convinced to cite this paper.
Query 2: As a second major revision is directed to the animal experiments. First of all, sample number (number per experimental group) used was not mentioned in M&M section. Authors are stating that „defect filling was significantly higher in the animals with granular bone substitutes than in the control as shown in histomorphometry analysis“, but it is not elaborated in M&M how histomorphometry was performed and how was newly formed bone quantified? Also, microCT analysis would be a great option for a 3D quantification of newly formed bone. If there is no option for microCT analysis, I would suggest 2D histomorphometry analysis for newly formed bone described in Stokovic et al., Autologous bone graft substitute containing rhBMP6 within autologous blood coagulum and synthetic ceramics of different particle size determines the quantity and structural pattern of bone formed in a rat subcutaneous assay (doi: 10.1016/j.bone.2020.115654).
Answer: We appreciate your important comments. We would like to explain that the histomorphometry and newly formed bone quantification was done by using the micro-CT analysis results and software. The micro-CT results are already published while reporting the preliminary results. Same materials has been used for the clinical trial at present study.
More details can be found easily at given doi. https://doi.org/10.1016/j.msec.2020.110694

Reviewer 2 Report
The work contains some interesting data related to the application of a multichannel bone substitute for bone regeneration, adopting some different specific pathological cases.
Anyway, it lacks important aspects that should be included by authors.
Firstly, introduction section should be modified by including some aspect related to the proposed methodology and also should focus on innovative processes (i.e. CAD/CAM technologies or Additive Manufacturing – AM – techniques) for 3D structures fabrication with specific and planned morphological and functional features. In this direction, the authors should also mention the advantages related to the adopted method if compared to other and advanced AM technologies.
An appropriate characterization of the proposed 3D structures in terms of morphology and mechanical features should be provided by the authors. Scanning electron microscopy (SEM) measurements – mentioned in the text, but not provided in the results section - should be employed thus obtaining a deep characterization of the materials adopted for the purpose, also focusing on pore shape and size of the entire 3D structure as well as providing information regarding eventual shape deformation and micro-/nano-porosity that should be obtained because of the employed fabrication method.
In vivo studies on early bone regeneration should be provided in a deeper and high-quality manner also adopting and comparing other kind of immunostaining ( Col I, Alp, Bmp-2, Ocn, Opn, etc.).
References should be properly revised and updated.
Author Response
Query 3: Firstly, introduction section should be modified by including some aspect related to the proposed methodology and also should focus on innovative processes (i.e. CAD/CAM technologies or Additive Manufacturing – AM – techniques) for 3D structures fabrication with specific and planned morphological and functional features. In this direction, the authors should also mention the advantages related to the adopted method if compared to other and advanced AM technologies.
Answer: We appreciate your comment. You raised an important point. We agree with your opinion. We are agreed to reviewer and now edited the introduction part with brief explanation for advantages of the adopted method. We accept that your point and added it in the introduction in the revised text; “However, fabricating porous bone graft with sufficient mechanical strength is challenging because the scaffold’s porosity compromises strength. To overcome this problem, mul-ti-pass extrusion fabrication has been used to create calcium phosphate-based multi-channel porous granules (MCG), which allows sufficient control of pore size, shape and mechanical properties. This technology also minimizes the inverse relationship between compressive strength and porosity, which is a limitation with many of the existing syn-thetic bone graft materials.” (9th line of 2nd paragraph in the Introduction)
Query 4: An appropriate characterization of the proposed 3D structures in terms of morphology and mechanical features should be provided by the authors. Scanning electron microscopy (SEM) measurements – mentioned in the text, but not provided in the results section - should be employed thus obtaining a deep characterization of the materials adopted for the purpose, also focusing on pore shape and size of the entire 3D structure as well as providing information regarding eventual shape deformation and micro-/nano-porosity that should be obtained because of the employed fabrication method.
Answer: Authors appreciate the careful observation of reviewer. Authors have already provided information on porosity at section 3.1 and also Figure 2.The non-described part has now been removed from the text part. Detailed microstructural analysis and pore size analysis and porosity data have already published and can be easily found at DOI: 10.1177/0885328218768605.
Query 5: In vivo studies on early bone regeneration should be provided in a deeper and high-quality manner also adopting and comparing other kind of immunostaining (Col I, Alp, Bmp-2, OCN, OPN, etc.).
Answer: Authors have already provided the results in a deeper and high-quality manner. These experiments earlier and are published. OCN and OPN data are reported in our previous publication, unfortunately we don’t have good results for Col I, ALP and BMP2 and at present we are unable to perform too.
Results are available at https://doi.org/10.1016/j.msec.2020.110694
Query 6: References should be properly revised and updated.
Answer: Authors have tried their level best to update the references.

Reviewer 3 Report
It seems to me a well-written paper and an interesting topic for publication, in order to improve the understanding of the final results and make reading easier, I suggested checking the following:
Line 128 (in vivo characterization section), does not clarify the number of biomodels used for each group.
Line 298, in figure 3, the legend seems to describe only some of the items and that is a bit confusing, for example if you want to explain items a, b and c it is suggested to put (a-c), writing (a,c) it suggests that you are only explaining 2 photographs, a and c, in that case, there are some photographs without description.
Finally, the discussion part explains in detail about the results, in addition to justifying each of the analyzes and essays of the work, it seems to me an explanatory way of approaching the discussion, but I have felt a lack of comparison of this information with that of other similar works or related to the area of bone regeneration. In this same section, more recent, novel and relevant bibliographical references could be added, since only one is from 2018, the rest are more than 10 years old.
Author Response
Query 7: Line 128 (in vivo characterization section), does not clarify the number of biomodels used for each group.
Answer: Authors apologized for missing this information. We have used standard n=3 for each group. We add the sentence. “For each set of samples, n=3 were used.” (5th line of 4th paragraph in the Material and Methods)
Query 8: Line 298, in figure 3, the legend seems to describe only some of the items and that is a bit confusing, for example if you want to explain items a, b and c it is suggested to put (a-c), writing (a,c) it suggests that you are only explaining 2 photographs, a and c, in that case, there are some photographs without description.
Answer: Authors apologized for their mistake, we have now corrected this part at the figure caption and highlighted in yellow. We revise the legend “H&E stained tissue section of granular bone substitute implanted bone areas. (a,b,c) Bone for-mation after 1 month of implantation with 1mm granule. (d,e,f) bone formation after 2 months of implantation. Legend: BS(Bone substitute), HB (Host Bone), NB( New Bone)”
Query 9: Finally, the discussion part explains in detail about the results, in addition to justifying each of the analyzes and essays of the work, it seems to me an explanatory way of approaching the discussion, but I have felt a lack of comparison of this information with that of other similar works or related to the area of bone regeneration. In this same section, more recent, novel and relevant bibliographical references could be added, since only one is from 2018, the rest are more than 10 years old.
Answer:
Authors are considering the reviewers suggestion and we have now tried our best to add latest references if any.
